# The Role of Transthoracic Echocardiography for Assessment of Mortality in Patients with Carcinoid Heart Disease Undergoing Valve Replacement

**DOI:** 10.3390/cancers15061875

**Published:** 2023-03-21

**Authors:** Abigail Brooke, Sasha Porter-Bent, James Hodson, Raheel Ahmad, Tessa Oelofse, Harjot Singh, Tahir Shah, Ahmed Ashoub, Stephen Rooney, Richard P. Steeds

**Affiliations:** 1Department of Cardiology, University Hospitals Birmingham (Queen Elizabeth) NHS Hospitals Foundation Trust, Birmingham B15 2TH, UK; 2Institute of Cardiovascular Sciences, University of Birmingham, University Hospitals Birmingham (Queen Elizabeth) NHS Hospitals Foundation Trust, Birmingham B15 2TH, UK; 3Institute of Translational Medicine, University Hospitals Birmingham (Queen Elizabeth) NHS Hospitals Foundation Trust, Birmingham B15 2TH, UK; 4Department of Cardiac Anaesthesia and Intensive Care, University Hospitals Birmingham (Queen Elizabeth) NHS Hospitals Foundation Trust, Birmingham B15 2TH, UK; 5Birmingham Neuroendocrine Tumour Centre, University Hospitals Birmingham (Queen Elizabeth) NHS Hospitals Foundation Trust, Birmingham B15 2TH, UK; 6Department of Cardiothoracic Surgery, University Hospitals Birmingham (Queen Elizabeth) NHS Hospitals Foundation Trust, Birmingham B15 2TH, UK

**Keywords:** neuroendocrine tumour, carcinoid heart disease, valvular heart disease, transthoracic echocardiography, right ventricle, cardiothoracic surgery

## Abstract

**Simple Summary:**

A proportion of patients with neuroendocrine tumours develop carcinoid syndrome and subsequent carcinoid heart disease (CHD). Valve replacement is indicated for patients with CHD when there is severe regurgitation and symptoms or evidence of right ventricular (RV) failure. The risk of 30-day mortality is high, yet the prognosis in those managed conservatively is poor. Consensus recommendations are that surgery is limited to those expected to live more than 12 months, but factors known to predict outcome are lacking. The aim of our retrospective study was to identify preoperative echocardiographic parameters that may be associated with prognosis. In our cohort of 49 patients with CHD undergoing valve surgery, we show a significant association between preoperative RV size and outcome, with one-year mortality rates of 57% vs. 33% for those with severe RV dilatation vs. normal RV size. This raises the question of whether surgery should be considered earlier, prior to RV dilatation.

**Abstract:**

Patients with carcinoid heart disease (CHD) are referred for valve replacement if they have severe symptomatic disease or evidence of right ventricular (RV) failure and an anticipated survival of at least 12 months. Data are lacking, however, on the role of transthoracic echocardiography in predicting outcomes. We carried out a retrospective, single-centre cohort study of patients with a biopsy-confirmed neuroendocrine tumour (NET) and CHD undergoing valve replacement for severe valve disease and symptoms of right heart failure. The aim was to identify factors associated with postoperative mortality, both within one year of surgery and during long-term follow-up. Of 88 patients with NET, 49 were treated surgically (mean age: 64.4 ± 7.6 years; 55% male), of whom 48 had a bioprosthetic tricuspid valve replacement for severe tricuspid regurgitation; 39 patients had a pulmonary valve replacement. Over a median potential follow-up of 96 months (interquartile range: 56–125), there were 37 deaths, with 30-day and one-year mortality of 14% (*n* = 7) and 39% (*n* = 19), respectively. A significant relationship between RV size and one-year mortality was observed, with 57% of those with severe RV dilatation dying within a year of surgery, compared to 33% in those with normal RV size (*p* = 0.039). This difference remained significant in the time-to-event analysis of long-term survival (*p* = 0.008). RV size was found to reduce significantly with surgery (*p* < 0.001). Those with persisting RV dilatation (*p* = 0.007) or worse RV function (*p* = 0.001) on postoperative echocardiography had significantly shorter long-term survival. In this single-centre retrospective study of patients undergoing surgery for CHD, increasingly severe RV dilatation on preoperative echocardiography predicted adverse outcomes, yielding a doubling of the one-year mortality rate relative to normal RV size. These data support the possibility that early surgery might deliver greater long-term benefits in this patient cohort.

## 1. Introduction

The prevalence of neuroendocrine tumours (NETs) is increasing [1,2,3], with a corresponding growth in the prevalence of carcinoid syndrome (CS) [4]. CS is a paraneoplastic syndrome that arises due to the release of vasoactive substances such as serotonin (5-hydroxytryptamine), prostaglandins and kallikrein into the systemic circulation, causing characteristic effects including flushing, diarrhoea, and abdominal pain [5,6]. It is unclear why, but in approximately 20% of patients with CS, carcinoid heart disease (CHD) develops, with progressive leaflet thickening, retraction and reduced mobility predominantly affecting the tricuspid and pulmonary valves [7]. Those patients with CS who develop CHD have a worse prognosis, with average survival half that of NET patients without CHD [8,9]. There are currently no medical therapies that can prevent the onset or retard the progression of CHD, and insidious valve destruction ultimately results in right atrial and ventricular dilatation and dysfunction, leading to premature heart failure and death [10]. Current therapy is restricted to open surgical valve replacement, and although perioperative mortality has fallen in highly specialised centres over time, 30-day mortality remains at around 9% [11]. There are currently no randomised controlled trials comparing surgery with conservative management, but studies have demonstrated the use of medical therapy alone to confer a poor prognosis, with median survival limited to one year in those with severe valve disease who are treated with medical therapy alone [12]. In patients who proceed to valve replacement, the safety profile is reasonable [12,13,14,15,16,17,18,19,20]. A meta-analysis of nine cohort studies reported an average survival of 69% at one year, with evidence that this rate is increasing due to better patient selection, improved surgical techniques and greater experience [11]. Whilst risk factors such as age over 70 years, advanced symptoms and active CS are associated with worse outcomes [21,22], by consensus, it is recommended that patients referred for cardiac surgery should either be symptomatic or have evidence of right heart failure and have an anticipated postoperative survival of at least 12-months [22]. The aim of this study is to investigate the associations between parameters measured on preoperative echocardiography and postoperative survival, both at one year and over the long term, in order to identify potential predictors of prognosis.

## 2. Materials and Methods

This was a retrospective observational cohort study, including patients referred for cardiology review between January 2005 and July 2019 at the Queen Elizabeth Hospital, University Hospitals Birmingham (UHB) NHS Foundation Trust, United Kingdom, with follow-up to July 2021. UHB is a European Neuroendocrine Tumour Society certified Centre of Excellence for NET and has a specialist team working in the field, with multidisciplinary, multispecialty one-stop clinics for CHD [23]. Patients were included in this study following referral for surgery based on the following criteria: (1) NET and CS diagnosis confirmed on biopsy, history and urinary 5-hydroxy-indole acetic acid (5-HIAA); (2) severe valvular dysfunction and (3) exercise-limiting symptoms or evidence of right heart failure. Patients were identified from databases maintained by the Cardiology and Cardiothoracic Surgical teams as part of routine practice.

### 2.1. Echocardiography

Transthoracic echocardiography (TTE) was performed using Philips IE33 and EPIQ machines by experienced, accredited sonographers, according to the Minimum Dataset of the British Society of Echocardiography [24]. CHD was diagnosed by the presence of characteristic thickening and reduced leaflet excursion with or without leaflet retraction, with associated evidence of regurgitation or stenosis on echocardiography. The severity of valvular regurgitation was graded according to multiparametric assessment in accordance with European Association of Cardiovascular Imaging guidelines, which included mild, moderate and severe grading but not massive or torrential [25]. Measurements of right and left-sided chamber dimensions and function were made in accordance with American Society of Echocardiography guidelines [26]. In brief, a qualitative assessment of right ventricular (RV) size was performed from the apical four-chamber view and considered normal if the right ventricle was smaller than that of the left ventricle (LV). In cases of mild enlargement, the RV cavity area was similar to that of the LV, but the LV was apex-forming. In the case of moderate enlargement, the RV cavity area was similar to that of the LV and shared the apex of the heart. In the case of severe enlargement, the RV cavity area exceeded that of the LV, and the right ventricle was apex forming. Quantitative assessment of RV size was performed in a non-foreshortened apical four-chamber view, oriented to obtain the maximum RV dimension, and basal RV diameter was measured at end-diastole (RVIDd, cm). RV function was assessed both qualitatively and quantitatively. The qualitative assessment involved visual assessment by experienced accredited echocardiographers dedicated to the CHD clinic, and quantitative measurements were carried out using tricuspid annular plane systolic excursion (TAPSE, cm), fractional area change (FAC, %) and tissue Doppler imaging of tricuspid annular velocity (RV S wave, cm/s). Linear measurements of LV internal dimension in diastole (LVIDd, cm) and systole (LVIDs, cm) were made from the parasternal long-axis acoustic window at the level of the LV minor axis, approximately at the mitral valve leaflet tips. Left ventricular ejection fraction (LVEF, %) was measured using the 2D biplane method of discs (modified Simpson’s rule) from the apical four and apical two-chamber views. Partition values into normal, mild, moderate and severe dilatation; and normal, mild, moderate and severe impairment were taken from the Recommendations for Chamber Quantification [26].

Transoesophageal echocardiography (TOE) was performed in all cases with the aim of providing high-quality imaging of the pulmonary valve and delineating any left-sided valve involvement [27].

All patients had an agitated saline contrast TTE for detection of a patent foramen ovale (PFO) following a standard protocol as previously described [28].

### 2.2. Cardiac Catheterisation

All patients referred for surgery underwent coronary angiography to assess for the presence of ischaemic heart disease and right heart catheterisation for assessment of right atrial (RA) pressure, RV pressure, mean pulmonary artery pressure (PAP) and pulmonary capillary wedge pressure (PCWP).

### 2.3. Data Collection

Initially, all patients undergoing surgery that met the inclusion criteria of the study were identified. For these, data for demographic, disease-related and operative factors, as well as dates of surgery and death (where applicable), were extracted from the UHB in-house clinical noting system ‘Clinical Portal’. The final preoperative TTE performed prior to the operation date and the first postoperative TTE (within 6 months of surgery) were then identified, and data for the parameters of interest were collected from the TTE reporting system Intellispace Cardiovascular (Philips).

### 2.4. Ethics

This was an observational, retrospective study limited to secondary use of information previously collected in the course of normal care (without an intention to use it for research at the time of collection) and is therefore excluded from ethical review according to the UK Health Research Authority decision tool. It was approved by local clinical governance committees (CARMS-15368) and conformed to the principles of Good Clinical Practice. Once collated, patient data were pseudonymised, with each patient assigned a random identifier and stored on a password-protected database that could only be accessed by direct members of the research team.

### 2.5. Statistical Methods

The distributions of continuous variables were assessed graphically, with normally distributed variables being summarised using mean ± standard deviation (SD) and medians and interquartile ranges (IQRs) used otherwise. Associations between TTE parameters and one-year mortality were then assessed using Mann–Whitney U tests for continuous or ordinal variables, with Fisher’s exact tests used for nominal variables.

The analysis of echocardiography parameters was then repeated using a time-to-event approach to assess associations with long-term survival. Continuous variables were individually entered into univariable Cox regression models. Nominal variables were analysed using two different approaches. Initially, these were treated as continuous and entered as covariates into Cox regression models. To quantify the difference between categories, the models were also repeated with the variables treated as nominal. Associations with survival were then visualised using Kaplan-Meier curves, with continuous variables divided into groups based on the tertiles of the distribution.

Comparisons between pre- and postoperative echocardiography parameters were then performed using Wilcoxon’s signed rank tests. All analyses were performed using IBM SPSS 24 (IBM Corp. Armonk, NY, USA), with *p* < 0.05 deemed to be indicative of statistical significance throughout.

## 3. Results

### 3.1. Demographics

Of 88 patients seen within the multidisciplinary NET-CHD clinic, 49 (56%) were treated surgically and so were included in the study. These patients had a mean age at the preoperative assessment of 64.4 ± 7.6 years, and 55% were male. Further demographics, comorbidities and functional status of the cohort are reported in Table 1. Preoperative TTE was performed a median of 78 days (IQR: 46–130) prior to surgery, the results of which are summarised in Table 2. In each case, surgery involved the implantation of a bioprosthetic valve, and no mechanical valve replacement or repair procedures were performed. Bioprostheses implanted were Medtronic Hancock II, Edwards Perimount or St Jude Epic valves. Of the 49 patients; 8 underwent isolated tricuspid valve replacement (TVR), 35 had TVR and concomitant pulmonary valve replacement (PVR), 2 had their tricuspid, pulmonary and aortic valves replaced, 1 underwent mitral and aortic valve replacement (AVR), 1 had TVR and AVR, and 2 patients had all four valves replaced.

In total, a PFO was closed at the time of surgery in 26% (12/46). No patient had evidence of pulmonary hypertension on preoperative cardiac catheterisation (Appendix A); five patients also underwent concomitant coronary artery bypass grafting (CABG). Following surgery, the median length of hospital stay was 14 days (IQR 10–18). Epicardial pacing leads (PPMs) were implanted in 48% (23/48) due to concerns regarding postoperative cardiac conduction.

### 3.2. Associations between Preoperative Variables and Postoperative Survival

From surgery to the time of data collection, the median potential follow-up time for the cohort was 96 months (IQR: 56–125). There were a total of 37 deaths, with 30-day and one-year mortality rates of 14% (*N* = 7) and 39% (*N* = 19), respectively, and a median survival time of 27 months (Figure 1). Associations between one-year mortality and a range of patient and operative factors were then assessed, with no significant relationships being observed (Table 1). There were no significant relationships observed between comorbidities and one-year mortality. Of the preoperative TTE parameters considered (Table 2), a significant relationship between RV size and one-year mortality was observed, with this being severely dilated in 68% of those that died within a year of surgery, compared to 33% of those that survived (*p* = 0.039). None of the other factors considered were found to be significant in this analysis.

The analyses of associations between preoperative TTE and postoperative survival were then repeated using a time-to-event approach (Table 3). Preoperative RV size remained a significant predictor of survival in this analysis (Figure 2A, *p* = 0.008), with an estimated survival rate at three years of 50% vs. 22% for those with normal RV size vs. severe RV dilatation. This analysis also identified LVEF as a significant predictor of long-term survival (Figure 2B, *p* = 0.038), with a three-year survival rate of 42% in those with normal LVEF, whilst the single patient with moderately impaired LVEF died 18 days after surgery.

### 3.3. Changes from Pre- to Early Postoperative Echocardiography

Postoperative echocardiography was performed in 94% (46/49) of patients at a median of 9 days (IQR: 6–15) after surgery. Comparisons between pre- and postoperative TTE parameters are reported in Table 4. RV size was found to reduce significantly with surgery (*p* < 0.001), with 65% being normal on the postoperative scan, compared to 12% preoperatively. However, there was also a worsening of RV function, with the number of patients classified as normal falling from 86% to 21% (*p* < 0.001). TAPSE (*p* = 0.003) and RV FAC (*p* < 0.001) were found to be significantly reduced after surgery, whilst TV Vmax (*p* = 0.019) and AV Vmax (*p* < 0.001) were significantly increased.

### 3.4. Associations between Postoperative TTE Parameters and Survival

The analysis of parameters was then repeated for the measurements made on postoperative echocardiography (Table 3). The RV size remained a significant predictor of survival in this analysis (Figure 3A), with an estimated survival rate at three years of 51% vs. 0% for those with normal RV size compared to those with severe RV dilatation (*p* = 0.007). Increasingly impaired postoperative RV function was also associated with significantly shorter survival (Figure 3B; *p* = 0.001), with all six patients with severe impairment dying within eight months of surgery. The single patient with moderately impaired postoperative LVEF also had significantly shorter survival than those with normal LVEF (*p* = 0.034), although there was no evidence of an ordinal trend across the categories of LVEF impairment (*p* = 0.664). Finally, higher RV FAC was found to be associated with significantly longer survival, with a hazard ratio of 0.94 per percentage point (*p* = 0.014). This is visualised in Figure 3C, with three-year survival rates of 13% vs. 54% in the lowest vs. highest tertile of RV FAC.

## 4. Discussion

Patients with metastatic NET, CS and severe carcinoid heart valve disease face a difficult decision with their clinicians regarding whether to undergo cardiothoracic surgery, usually to replace regurgitant tricuspid valves. There is a lack of evidence from randomised controlled trials of the benefit of surgery for primary TR of any aetiology, and there are no such data to support decision-making in CHD. Surgery is carried out in CHD with the primary objective of alleviating symptoms and the expectation of improved survival. There often remains a balance between the 9% risk of mortality within 30 days, substantial in-patient hospital stays (median 14 (IQR 10–18) days in our cohort), and the time taken to recover from median sternotomy, compared to a median life expectancy of 12 months in those treated conservatively. This is the first study to identify that increasing RV dilatation should be a factor to consider in terms of the likelihood of survival at one year, with a significant association between preoperative RV dilatation and mortality. This relationship separated out those with severe RV dilatation who had a mortality rate of 57% at one year, compared to 33% in those with normal RV size, an association that was replicated on time-to-event analyses. Whilst RV remodelling took place in the majority following TVR **+/−** PVR, there were small numbers of patients for whom severe RV dilatation or dysfunction persisted following surgery. These represented subgroups of patients with particularly poor prognoses.

Previous studies have indicated that the prognosis following surgery for CHD is adversely affected by older age > 70 years (comparison 69:55) and by advanced symptoms (comparison NYHA III:II) [17]. Our cohort was of a similar age (64.4 ± 7.6 years), but a smaller percentage had advanced symptoms (39% in NYHA III or IV compared to 70%), yet neither of these factors were found to be significant predictors of one-year mortality in our study. The reason for this difference is not clear but may reflect a smaller sample size, although older age and advanced symptomatology are factors that are considered in our multidisciplinary team discussions, and this may have biased their impact [23]. It is important to note that preoperative measures of RV function were not found to be significant predictors of long-term survival, whether using a categorical approach or using values from tricuspid annular plane systolic excursion (TAPSE), RV systolic tissue velocity (s’) or fractional area change (FAC). Although the cause of death was not available in our study, it is well established that early postoperative death within 30 days is generally due to RV failure [23]. This reflects the limitations of these standard measures in patients with severe TR, in whom volume-loading of the RV causes over-estimation of myocardial contractility. RV global longitudinal strain was not measured in our cohort, but better preoperative measures are needed to detect the myocyte stretch, hypertrophy and sarcomeric stiffening that occurs in response to chronic excessive preload and that ultimately leads to myocyte death, fibrosis and irreversible RV dysfunction [29]. It is not known whether cardiovascular magnetic resonance imaging will be of incremental value in predicting postoperative outcomes.

The optimal timing of surgery for severe primary TR has not been established. Current guidelines, however, suggest that ‘serial assessments of RV size and function might trigger consideration of corrective surgery in selected patients with severe primary TR when a pattern of continued deterioration can be established, and the surgical risk is considered acceptable [30]. Serial assessments were not available for most patients in our study, and although we recognise that careful, longitudinal assessment may lead to more timely intervention and potentially improved outcomes, the data in this study highlight an important practical issue in the management of CHD. In our cohort, 41/49 had more than one valve replaced, and 88% had NYHA II or more severe symptoms, reflecting our clinical experience that these patients present late in the course of their disease when cardiac involvement is advanced and intervention needs to be decided with minimal delay. Monitoring of patients at risk of CHD, i.e., those with active carcinoid syndrome or high levels of urinary 5-hydroxy-indole acetic acid, remains variable both in terms of the method of tracking cardiac involvement (N-terminal pro-B natriuretic peptide or echocardiography) and in frequency with which these are done (entry to service or six-monthly intervals) [31]. Until diagnosis and tracking of CHD become routine and reliable, plans for earlier intervention may be thwarted.

### Limitations

The results of this study need to be interpreted considering its limitations. The primary limitation was the potential for selection bias in patients undergoing surgical treatment. It is likely that some of the factors considered in the analysis were used when considering whether to offer surgery to patients. As such, assessment of these factors would have taken place in a biased cohort; hence the results would only be generalizable to situations where the selection criteria for surgical eligibility are consistent with those used in the current study. Secondly, data were only assessed for those patients who underwent surgery, not for those treated medically. As a result, for the factors considered, it was not possible to assess the relative improvement in outcomes after surgery compared to what would be expected with medical treatment. Thirdly, the cohort was relatively small, which limited the analyses that could be performed. Specifically, it was not deemed feasible to produce a reliable multivariable model to identify factors independently associated with survival. This was because such a model would either need to adjust for all potential factors of interest, leading to an increased risk of overfitting and limited statistical power, or adjust for a limited subset of factors, which would result in a high risk of residual confounding. The small sample size also resulted in low statistical power, meaning that only large effects were detectable and that potential predictors of patient outcomes may have been missed. For example, a post hoc power calculation of the Cox regression models with the observed sample size and event rate estimated a minimal detectable hazard ratio of 2.4 at 80% power and 5% alpha. Finally, this was a retrospective study using routinely collected clinical data, which was incomplete for some of the parameters considered, and inter- and intra-operator variability was not assessed. However, previous studies have reported acceptable reproducibility when assessing RV size on 2D echocardiography. For example, for the evaluation of RV basal dimensions from an RV focussed view, as used in our study, intraclass correlation coefficients (ICCs) of 0.94 (95% confidence interval 0.9–0.97) for intra-operator variability and 0.93 (95% CI 0.88–0.96) for inter-operator variability have been reported [32].

## 5. Conclusions

In this single-centre retrospective study of patients undergoing surgery for carcinoid heart disease, increasingly severe RV dilatation predicted adverse outcomes, with three-year survival in those with normal preoperative RV size being twice that of those with severe dilatation. These data support the possibility that early surgery might deliver greater long-term benefits in this patient cohort and that chronic exposure to excessive preload results in irreversible sub-clinical RV dysfunction that could remove the benefit of the intervention.

## Figures and Tables

**Figure 1 cancers-15-01875-f001:**
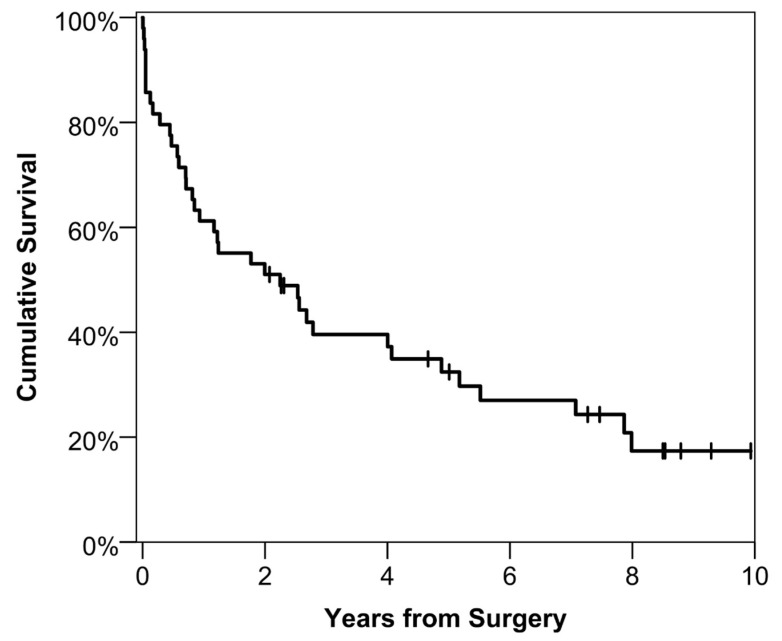
Kaplan–Meier Curve of Postoperative Survival.

**Figure 2 cancers-15-01875-f002:**
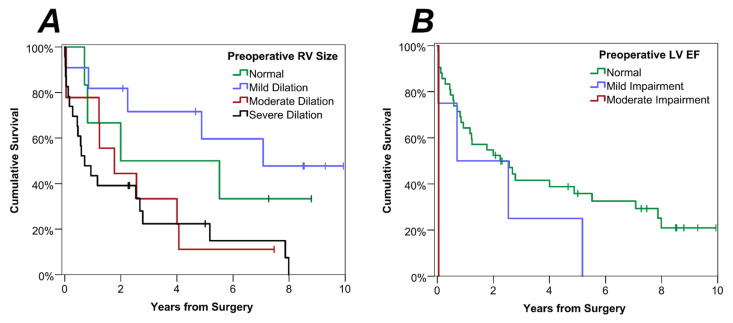
Kaplan-Meier Curves of Survival by Preoperative TTE Parameters. Separate curves were produced for subgroups defined by preoperative (**A**) RV size and (**B**) LVEF.

**Figure 3 cancers-15-01875-f003:**
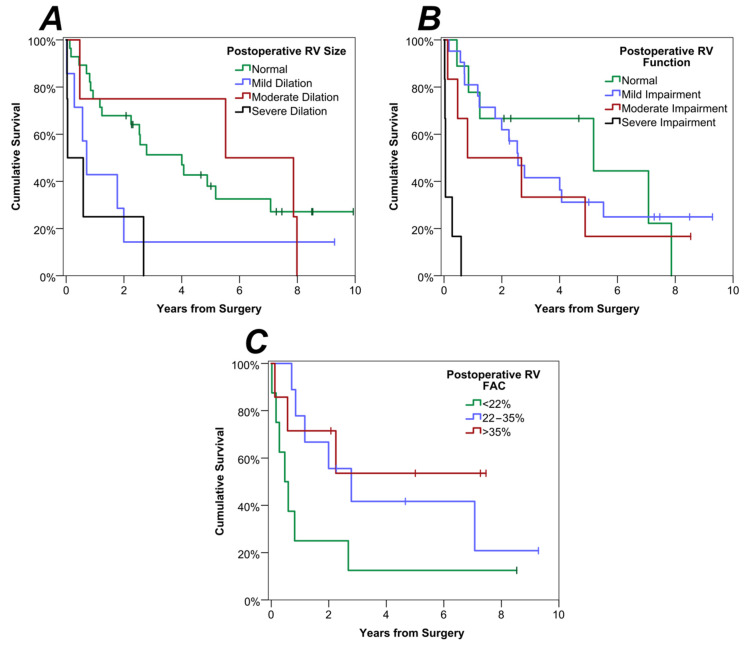
Kaplan-Meier Curves of Survival by Postoperative TTE Parameters. For the plot of postoperative RV FAC, patients were divided into three groups, based on tertiles of the distribution, to visualise the association with postoperative survival. Separate curves were produced for subgroups defined by postoperative (**A**) RV size and (**B**) RV function. In (**C**), patients were divided into three subgroups, based on tertiles of the distribution of postoperative RV FAC.

**Table 1 cancers-15-01875-t001:** Cohort Characteristics and Associations with One-Year Mortality.

	Overall		One-Year Postoperative Mortality				
			No		Yes		*p*-Value
	N	Statistic	N	Statistic	N	Statistic	
**Preoperative Factors**							
Age at Assessment (Years)	49	64.4 ± 7.6	30	64.7 ± 7.8	19	64.1 ± 7.5	0.629
Gender (% Female)	49	22 (45%)	30	15 (50%)	19	7 (37%)	0.395
BMI (kg/m^2^)	49	23.6 (21.9–26.9)	30	23.7 (22.2–26.3)	19	23.6 (20.0–29.9)	0.886
Active Weight Loss	27	12 (44%)	16	7 (44%)	11	5 (45%)	1.000
Diabetes	40	6 (15%)	25	3 (12%)	15	3 (20%)	0.645
Hypertension	40	23 (58%)	25	14 (56%)	15	9 (60%)	1.000
Smoking Status	40		25		15		1.000
*Non-*		34 (85%)		21 (84%)		13 (87%)	
*Ex-*		4 (10%)		3 (12%)		1 (7%)	
*Current*		2 (5%)		1 (4%)		1 (7%)	
NYHA Class	41		26		15		0.468 *
*I*		5 (12%)		2 (8%)		3 (20%)	
*II*		20 (49%)		13 (50%)		7 (47%)	
*III*		15 (37%)		11 (42%)		4 (27%)	
*IV*		1 (2%)		0 (0%)		1 (7%)	
5HIAA Excretion (u/mol/24 h)	35	719 (341–1222)	22	748 (389–1222)	13	719 (199–1200)	0.585
EuroScore II	49	2.43 (1.41–3.16)	30	2.62 (1.41–3.12)	19	2.25 (1.30–4.83)	0.829
**Peri-/Postoperative Factors**							
CABG	46	5 (11%)	29	3 (10%)	17	2 (12%)	1.000
PFO Closure	46	12 (26%)	29	7 (24%)	17	5 (29%)	0.737
PPM Inserted	48	23 (48%)	30	15 (50%)	18	8 (44%)	0.772
Pulmonary Valve Replacement	49	39 (80%)	30	25 (83%)	19	14 (74%)	0.414
Postoperative Length of Stay (Days)	47	14 (10–18)	29	14 (9–17)	18	15 (12–20)	0.398

Continuous data are reported as median (interquartile range) or as mean ± SD, as appropriate, with *p*-values from Mann-Whitney U tests. Nominal data are reported as N (column %), with *p*-values from Fisher’s exact tests, unless stated otherwise. Bold *p*-Values are significant at *p* < 0.05. * *p*-Value from Mann-Whitney U test, as the factor is ordinal.

**Table 2 cancers-15-01875-t002:** Association Between Preoperative Echocardiography and One-Year Mortality.

	Overall		One-Year Postoperative Mortality				
			No		Yes		*p*-Value
	N	Statistic	N	Statistic	N	Statistic	
TTE to Surgery (Days)	48	78 (46–130)	29	87 (47–132)	19	67 (45–127)	0.689
RV Base Diameter (cm)	44	4.5 ± 0.9	26	4.4 ± 0.9	18	4.7 ± 0.9	0.075
RV Size	49		30		19		**0.039 ***
Normal		6 (12%)		4 (13%)		2 (11%)	
Mild Dilation		11 (22%)		9 (30%)		2 (11%)	
Moderate Dilation		9 (18%)		7 (23%)		2 (11%)	
Severe Dilation		23 (47%)		10 (33%)		13 (68%)	
RV Function	49		30		19		0.840 *
Normal		42 (86%)		26 (87%)		16 (84%)	
Mild Impairment		4 (8%)		2 (7%)		2 (11%)	
Moderate Impairment		3 (6%)		2 (7%)		1 (5%)	
Severe Impairment		0 (0%)		0 (0%)		0 (0%)	
TAPSE (cm)	42	2.4 (2.0–2.7)	24	2.3 (2.1–2.8)	18	2.4 (2.0–2.6)	0.779
RV FAC (%)	34	52 (42–55)	19	53 (49–56)	15	45 (37–53)	0.165
RV-PA Coupling (mm/mmHG)	37	0.74 (0.59–0.94)	21	0.76 (0.63–0.96)	16	0.69 (0.58–0.92)	0.490
RV S Wave (cm/s)	27	14.8 ± 3.5	12	13.8 ± 2.2	15	15.5 ± 4.1	0.204
LV Size	48		29		19		1.000 *
Normal		48 (100%)		29 (100%)		19 (100%)	
Mild Dilation		0 (0%)		0 (0%)		0 (0%)	
Moderate Dilation		0 (0%)		0 (0%)		0 (0%)	
Severe Dilation		0 (0%)		0 (0%)		0 (0%)	
LVIDd (cm)	42	4.0 ± 0.6	25	3.9 ± 0.5	17	4.1 ± 0.6	0.369
LVIDs (cm)	41	2.7 ± 0.6	25	2.6 ± 0.6	16	2.8 ± 0.7	0.391
EF (%)	46	59.5 ± 5.4	28	60.1 ± 4.6	18	58.7 ± 6.5	0.505
LV EF	47		29		18		0.278 *
Normal		42 (89%)		27 (93%)		15 (83%)	
Mild Impairment		4 (9%)		2 (7%)		2 (11%)	
Moderate Impairment		1 (2%)		0 (0%)		1 (6%)	
Severe Impairment		0 (0%)		0 (0%)		0 (0%)	
TV Vmax (m/s)	25	1.3 ± 0.3	14	1.3 ± 0.3	11	1.3 ± 0.3	0.956
TR Vmax (m/s)	40	2.6 ± 0.6	24	2.6 ± 0.7	16	2.5 ± 0.6	0.480
PV Vmax (m/s)	44	1.6 ± 0.6	25	1.7 ± 0.7	19	1.5 ± 0.6	0.469
AV Vmax (m/s)	46	1.1 (0.9–1.4)	27	1.1 (0.9–1.4)	19	1.1 (0.9–1.3)	0.728

Continuous data are reported as median (interquartile range) or as mean ± SD, as appropriate, with *p*-values from Mann-Whitney U tests. Nominal data are reported as N (column %), with *p*-values from Fisher’s exact tests, unless stated otherwise. Bold *p*-values are significant at *p* < 0.05. * *p*-Value from Mann-Whitney U test, as the factor is ordinal.

**Table 3 cancers-15-01875-t003:** Association Between TTE Parameters and Long-Term Survival.

		Preoperative TTE			Postoperative TTE	
	N	HR (95% CI)	*p*-Value	N	HR (95% CI)	*p*-Value
TAPSE (per cm)	42	0.85 (0.50–1.46)	0.563	19	1.73 (0.93–3.20)	0.082
RV FAC (per pp)	34	0.99 (0.96–1.02)	0.577	24	0.94 (0.90–0.99)	**0.014**
RV-PA Coupling (per mm/mmHG)	37	0.97 (0.51–1.83)	0.923	-	NA **	-
RV S Wave (per cm/s)	27	1.09 (0.95–1.27)	0.229	-	NA **	-
LVIDd (per cm)	42	1.14 (0.61–2.13)	0.687	44	0.85 (0.43–1.67)	0.632
LVIDs (per cm)	41	1.36 (0.74–2.51)	0.323	42	1.04 (0.52–2.06)	0.919
TV Vmax (per m/s)	25	2.05 (0.43–9.88)	0.371	40	0.82 (0.22–2.96)	0.756
TR Vmax (per m/s)	40	0.85 (0.50–1.47)	0.570	-	NA **	-
PV Vmax (per m/s)	44	0.95 (0.56–1.63)	0.864	39	1.52 (0.81–2.84)	0.191
AV Vmax (per m/s)	46	1.01 (0.44–2.32)	0.983	32	2.11 (0.77–5.77)	0.146
RV Base Diameter (per cm)	44	1.27 (0.86–1.87)	0.228	37	1.92 (0.88–4.19)	0.100
RV Size	49	1.57 (1.13–2.18) *	**0.008 ***	43	1.40 (1.00–1.95) *	0.052 *
Normal	6	1	-	28	1	-
Mild Dilation	11	0.58 (0.15–2.16)	0.414	7	2.17 (0.85–5.54)	0.104
Moderate Dilation	9	1.98 (0.59–6.66)	0.272	4	1.05 (0.35–3.16)	0.927
Severe Dilation	23	2.37 (0.80–7.01)	0.118	4	4.65 (1.52–14.2)	**0.007**
RV Function	49	1.11 (0.63–1.95) *	0.727 *	42	2.21 (1.38–3.54) *	**0.001 ***
Normal	42	1	-	9	1	-
Mild Impairment	4	0.86 (0.26–2.89)	0.809	21	1.12 (0.43–2.89)	0.816
Moderate Impairment	3	1.39 (0.42–4.60)	0.588	6	1.52 (0.46–5.02)	0.487
Severe Impairment	0	-	-	6	22.92 (5.48–96.0)	**<0.001**
LV EF	47	2.35 (1.05–5.28) *	**0.038 ***	46	1.10 (0.72–1.69) *	0.664 *
Normal	42	1	-	38	1	-
Mild Impairment	4	1.93 (0.67–5.55)	0.220	4	0.82 (0.25–2.71)	0.740
Moderate Impairment	1	9.52 (1.11–81.9)	**0.040**	1	10.80 (1.20–97.0)	**0.034**
Severe Impairment	0	-	-	3	1.15 (0.27–4.89)	0.845

Results are from univariable Cox regression models. For continuous variables, hazard ratios are reported per one-unit increase in the variable. For ordinal variables, two models were produced, the first of which treated the variable as a continuous covariate, with the hazard ratio reported per one category increase. The second model treated the variables as nominal and calculated hazard ratios for each category relative to the “normal” category. * From a model treating the variable as continuous, hazard ratio is per one-category increase. ** Excluded from analysis since postoperative measurements were available for less than ten patients. Bold *p*-Values are significant at *p* < 0.05. HR = hazard ratio, pp = percentage point.

**Table 4 cancers-15-01875-t004:** Changes in TTE Parameters Between the Pre- and Postoperative Scans.

		Timing of TTE		Direction of Change *			
	N	Pre-Operative	Post-Operative	Reduced	No Change	Increased	*p*-Value
RV Base Diameter (cm)	35	4.6 ± 0.9	3.6 ± 0.6	29	2	4	**<0.001**
RV Size	43			33	8	2	**<0.001**
Normal		5 (12%)	28 (65%)				
Mild Dilation		10 (23%)	7 (16%)				
Moderate Dilation		6 (14%)	4 (9%)				
Severe Dilation		22 (51%)	4 (9%)				
RV Function	42			2	11	29	**<0.001**
Normal		36 (86%)	9 (21%)				
Mild Impairment		3 (7%)	21 (50%)				
Moderate Impairment		3 (7%)	6 (14%)				
Severe Impairment		0 (0%)	6 (14%)				
TAPSE (cm)	17	2.3 (2.1–2.6)	0.9 (0.7–1.4)	16	0	1	**0.003**
RV FAC (%)	24	51 (42–54)	34 (20–36)	20	1	3	**<0.001**
RV-PA Coupling (mm/mmHG) **	-	-	-	-	-	-	-
RV S wave (cm/s) **	-	-	-	-	-	-	-
LV Size	45			0	43	2	0.500
Normal		45 (100%)	43 (96%)				
Mild Dilation		0 (0%)	2 (4%)				
Moderate Dilation		0 (0%)	0 (0%)				
Severe Dilation		0 (0%)	0 (0%)				
LVIDd (cm)	39	4.0 ± 0.6	4.1 ± 0.6	14	4	21	0.249
LVIDs (cm)	37	2.7 ± 0.6	2.6 ± 0.6	15	3	19	0.882
EF (%)	43	59.4 ± 5.5	56.3 ± 12.5	18	12	13	0.127
LV EF	45			3	36	6	0.199
Normal		40 (89%)	38 (84%)				
Mild Impairment		4 (9%)	3 (7%)				
Moderate Impairment		1 (2%)	1 (2%)				
Severe Impairment		0 (0%)	3 (7%)				
TV Vmax (m/s)	21	1.3 ± 0.3	1.5 ± 0.2	3	1	17	**0.019**
TR Vmax (m/s) **	-	-	-	-	-	-	-
PV Vmax (m/s)	34	1.7 ± 0.6	1.9 ± 0.6	13	1	20	0.115
AV Vmax (m/s)	31	1.2 (1.0–1.4)	1.4 (1.2–1.8)	4	4	23	**<0.001**

Data are reported as N (column %), mean ± SD or median (IQR), as applicable. For each parameter, only those patients for whom values were recorded both pre- and postoperatively were included in the analysis. *p*-Values are from Wilcoxon’s signed ranks tests, and bold values are significant at *p* < 0.05. * The number of patients for whom values decreased/increased between the pre- and postoperative scans. ** Excluded from analysis since pairs of pre- and postoperative measurements were available for less than ten patients.

## Data Availability

The data presented in this study are available on request from the corresponding author.

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
