# Peer review of "The Role of Transthoracic Echocardiography for Assessment of Mortality in Patients with Carcinoid Heart Disease Undergoing Valve Replacement"

_cancers, 2023, doi:10.3390/cancers15061875_

Round 1

Reviewer 1 Report

The authors performed a retrospective study of 49 patients with neuroendocrine tumour and carcinoid disease requiring heart valve surgery and associated echocardiographic factors with post-operative survival. They identified right ventricular size as only predictive factor.

This is a very important manuscript, as data on patients with carcinoid disease is very rare. Furthermore, the decision to perform cardiac surgery may be especially difficult as some patients may have a limited prognosis due to the nature of the disease itself.

In my opinion, this is a well-written manuscript with adequate literature, written in good English. However, I would recommend to add baseline characteristics, such as age, comorbidities and progression of the underlying disease into the multivariable analysis.

Author Response

‘This is a very important manuscript, as data on patients with carcinoid disease is very rare. Furthermore, the decision to perform cardiac surgery may be especially difficult as some patients may have a limited prognosis due to the nature of the disease itself.

In my opinion, this is a well-written manuscript with adequate literature, written in good English. However, I would recommend to add baseline characteristics, such as age, comorbidities and progression of the underlying disease into the multivariable analysis.’

Thank you for the positive comments regarding our manuscript.

We agree with the reviewer that age, co-morbidities and data on progression (5HIAA) are important to consider in this cohort. As such, we have reported details of these baseline characteristics, along with univariable analyses of associations with one-year mortality in Table 1.

The manuscript does not currently include any multivariable analysis, with the results in Table 3 being from individual univariable Cox regression models, as stated in the footnote. Whilst we agree that a multivariable analysis would be useful, we made the decision not to perform such an analysis on account of the small sample size, with only N=37 deaths, which would be further reduced by exclusion of patients with missing data on factors of interest, since multivariable regression models are complete-cases analyses. As such, only three factors could be considered in a multivariable model, if observing the typical “one in ten” rule. Producing a multivariable model adjusting for so few factors would result in a high risk of under-adjustment leading to residual confounding, whilst adjusting for a greater number of factors would result in an excessive risk of overfitting and insufficient statistical power. Both of these possibilities would yield a model of questionable reliability, which may have resulted in inaccurate conclusions being drawn. As such, we have not performed a multivariable analysis, and have added a comment to the limitations section to address this.

Reviewer 2 Report

The authors submitted a research article in which they elucidated factors associated with post-operative mortality, both within one year of surgery and during long-term follow-up. This was a retrospective observational cohort study in which the authors included 88 patients with biopsy-confirmed neuroendocrine tumour and CHD undergoing valve replacement for severe valve disease and symptoms or right heart failure. The authors established that identify that increasing RV dilatation should be a factor to consider in terms of the likelihood of survival at one-year, with a significant association between pre-operative RV dilatation and mortality. Finally, the authors concluded that the data support the possibility that early surgery might deliver greater long-term benefits in this patient cohort. The results of the study seem to be impressive and I would like to congratulate the authors on them. However, there some comments which I wuld like to put forward.

1. The authors described demographics, but not comorbidities, which need to explain changes of cardiac hemodynemics along with tumor. Please, add thi information in the section "Results".

2. How long positive changes of cardiac function after surgical procedure preserve ?

Author Response

The authors submitted a research article in which they elucidated factors associated with post-operative mortality, both within one year of surgery and during long-term follow-up. This was a retrospective observational cohort study in which the authors included 88 patients with biopsy-confirmed neuroendocrine tumour and CHD undergoing valve replacement for severe valve disease and symptoms or right heart failure. The authors established that identify that increasing RV dilatation should be a factor to consider in terms of the likelihood of survival at one-year, with a significant association between pre-operative RV dilatation and mortality. Finally, the authors concluded that the data support the possibility that early surgery might deliver greater long-term benefits in this patient cohort. The results of the study seem to be impressive and I would like to congratulate the authors on them. However, there some comments which I would like to put forward.’

Thank you for the positive comments regarding our manuscript.

‘The authors described demographics, but not comorbidities, which need to explain changes of cardiac hemodynemics along with tumor. Please, add this information in the section "Results”.’

Associations between comorbidities (including diabetes, hypertension etc.) and one-year mortality are reported in Table 1, alongside the demographic factors. None of these were found to be significantly associated with one-year mortality; and we stated in the text that “no significant relationships were observed” whilst referencing this table. However, in light of your comment, we have added a clarification to the text stating that “there were no significant relationships observed between comorbidities and one-year mortality”.

‘How long positive changes of cardiac function after surgical procedure preserve?’

Whilst this is an interesting and relevant question, the current study was not designed to investigate this, and only included data from routinely performed pre- and post-operative TTEs. This is an area of research that we hope to explore in the future. This however, would require a different study design, in which repeated TTEs are prospectively performed in eligible patients. As such, this is outside of the scope of the present study and we are not able to comment.

Reviewer 3 Report

To:

Editorial Board

Cancers

Title: “The Role of Transthoracic Echocardiography for Assessment of Mortality in Patients with Carcinoid Heart Disease Undergoing Valve Replacement”

Dear Editor,

I read this paper and I think that:

-          The retrospective nature of this paper is a limitation of this study. This should be discussed in a dedicated limitation section. Please provide.

-          The reproducibility of the transthoracic evaluations should be considered and performed. Authors should evaluate inter and intraobserver variability coefficients for these measurements. Please provide.

-          The small sample size is a limitation of this paper. This should be discussed in a dedicated limitation section.

-          Are the authors able to provide a post-hoc sample size calculation? Please provide.

-          Authors can consider and discuss the paper from Tucci M et al. Biomedicines. 2021 Jul 3;9(7):774.

Author Response

‘The retrospective nature of this paper is a limitation of this study. This should be discussed in a dedicated limitation section. Please provide." AND "The small sample size is a limitation of this paper. This should be discussed in a dedicated limitation section.’

Whilst the limitations of the study were discussed in a paragraph at the end of the discussion section, we have now made this into a dedicated section, in light of your comment. We have also extended this to include the retrospective study design as well as the small sample size, as per your recommendations.

‘The reproducibility of the transthoracic evaluations should be considered and performed. Authors should evaluate inter and intra observer variability coefficients for these measurements. Please provide.’

This is an interesting point, and something that we acknowledge could add to the manuscript. All of the echocardiographers who performed TTEs on this cohort were accredited, and worked in the same department, suggesting that they should have been consistent. However, it is unfortunately not possible to reliably assess the inter-/intra-operator variability for several reasons. Primarily, this was a retrospective study that included TTEs performed over almost 15 years, with some of the earlier scans no longer being readily available to re-evaluate. In addition, some of these echocardiographers have now moved from our department, and so could not be included in any reproducibility analysis. As such, assessing reproducibility with an incomplete subset of the TTEs and echocardiographers may not give a reliable estimate of this. However, previous studies have shown high levels of reproducibility of TTE evaluations. For example, Genovese et al reported that, for measures of RV basal dimensions from an RV focussed view on 2D echocardiography, as used in our study, the intraclass correlation coefficient (ICC) was found to be 0.94 (95% confidence interval 0.9-0.97) for intra-operator variability and 0.93 (95% CI 0.88-0.96) inter-operator variability [Genovese et al, J Am Soc Echocardiogr. 2019 Apr;32(4):484-494]. We have commented on this in our limitations section.

"Are the authors able to provide a post-hoc sample size calculation? Please provide."

We have added a post-hoc power calculation of the survival analysis to the limitations section. To simplify the calculation, this was based on a log-rank test of a variable dichotomized by the median, and assumed the observed 10-year survival rate. This returned a minimal detectable hazard ratio of 2.4 at 80% power and 5% alpha. As mentioned in the paper, this is a relatively large effect size, and so will have resulted in an inflated false-negative rate, which is a limitation of the analysis.   

"Authors can consider and discuss the paper from Tucci M et al. Biomedicines. 2021 Jul 3;9(7):774."

Thank you for directing us to this informative and interesting paper. However we did not feel it was within the same scope of our manuscript, as it focuses on primary soft tissue sarcomas of the heart rather than the consequences of neuroendocrine tumours on cardiac valves.

Round 2

Reviewer 3 Report

authors well addressed previous comments. THe paper improved very much